# Polyhydride CeH$_9$ with an atomic-like hydrogen clathrate structure

Xin Li [1], Xiaoli Huang[1], Defang Duan[1,2], Chris J. Pickard [2], Di Zhou[1], Hui Xie[1], Quan Zhuang[1], Yanping Huang[1], Qiang Zhou[1], Bingbing Liu[1] & Tian Cui[1]

Compression of hydrogen-rich hydrides has been proposed as an alternative way to attain the atomic metallic hydrogen state or high-temperature superconductors. However, it remains a challenge to get access to these states by synthesizing novel polyhydrides with unusually high hydrogen-to-metal ratios. Here we synthesize a series of cerium (Ce) polyhydrides by a direct reaction of Ce and H$_2$ at high pressures. We discover that cerium polyhydride CeH$_9$, formed above 100 GPa, presents a three-dimensional hydrogen network composed of clathrate H$_{29}$ cages. The electron localization function together with band structure calculations elucidate the weak electron localization between H-H atoms and confirm its metallic character. By means of Ce atom doping, metallic hydrogen structure can be realized via the existence of CeH$_9$. Particularly, Ce atoms play a positive role to stabilize the sublattice of hydrogen cages similar to the recently discovered near-room-temperature lanthanum hydride superconductors.

[1] State Key Laboratory of Superhard Materials, College of Physics, Jilin University, Changchun 130012, China. [2] Department of Materials Science & Metallurgy, University of Cambridge, 27 Charles Babbage Road, Cambridge CB3 0FS, UK. Correspondence and requests for materials should be addressed to X.H. (email: huangxiaoli@jlu.edu.cn) or to T.C. (email: cuitian@jlu.edu.cn)

The realization of metallic hydrogen is highly desirable because of its attractive properties, such as high energy density, high-temperature superconductivity, and super-fluidity. Pressure has been recognized as one of the most straightforward and robust way of getting metallic hydrogen, where changes in bond strength and geometry are induced without the complication of chemical adjustments. Tremendous efforts have been devoted, but unfortunately, solid hydrogen requires very high pressures to get into a metallic state[1–6]. Recently, metallization of solid hydrogen at 495 GPa has been reported, but such high pressure is a challenge for further experimental exploration[7]. Systematic studies on polyhydrides have indicated that nonhydrogen atoms maybe provide a chemical precompression effect inside hydrogen-rich compounds, which may potentially reduce the pressure required to metallization[8]. Moreover, these polyhydrides also present intriguing high-temperature superconductivity theoretically, such as $H_3S$ with 203 K at 200 GPa[9,10], $CaH_6$ with 235 K at 150 GPa[11], $LaH_{10}$ with 280 K at 210 GPa[12], and $YH_{10}$ with 310 K at 250 GPa[13]. Only the predicted high $T_c$ exceeding 200 K of $H_3S$[14–16] and $LaH_{10}$[17,18] have been undoubtedly confirmed by experiments thus far. These discoveries have raised new prospects that polyhydrides could become a new family of high-temperature conventional superconductors, but the principal experimental challenge lies in the difficulty of synthesizing the novel hydrogen-rich compounds proposed by theory.

Until recently, polyhydrides were usually synthesized under high pressures combined with high-temperature condition. In these polyhydrides, hydrogen atoms exist in different forms, including unpaired H, $H_2$, and $H_3$ units, and extended H networks. For instance, $FeH_5$ prepared from a laser-heated diamond anvil cell (DAC) above 130 GPa have been proposed as a layered structure built of atomic hydrogen only[19]. $NaH_7$ was synthesized above 40 GPa and 2000 K, but it presents nonmetallic properties with $H_3^-$ and $H_2$ units processing higher-frequency vibrons in Raman spectrum[20]. Subsequent work turned to polyhydrides with three-dimensional hydrogen structure. These polyhydrides might be an analog of high-$T_c$ superconductor and the representative example was reported in a body-centered cubic phase of $LaH_{10}$ with clathrate $H_{32}$ cage[12], which was synthesized above 170 GPa and 1000 K. The surprising high $T_c$ near room temperature in lanthanum hydride is mainly attributed to an intriguing H clathrate structure with large H-derived electron density of states at Fermi level and the strong electron–phonon coupling related to H cages[21]. Then, in-depth research in rare earth metal hydrides is urgently needed, including successful synthesis of H clathrate structure and observation of superconductivity.

For these synthesized polyhydrides, the laser-heating treatment at sufficient pressure is the necessary condition of successful synthesis, overcoming the possible energy barrier. In contrast, to obtain polyhydrides only through the high-pressure method is more feasible to explore the promising properties. Here, we report a study of the prototypical Ce hydrides to find the one with atomic-like hydrogen sublattice and high-$T_c$ superconductor through cold-compression engineering. Among the Ce hydrides reported in the past decade, the studies of Ce hydrides are limited with a nonstoichiometric composition range ($CeH_x$, $x \leq 3$) by the combination of Ce in excess hydrogen[22,23]. In this work, the experimental syntheses of Ce polyhydrides have been successfully fulfilled under high pressures up to 159 GPa. Five stoichiometries of Ce polyhydrides are identified from experimental and theoretical results. Especially, we find that synthesized $CeH_9$ has unique clathrate structure with $H_{29}$ cages and exhibits a structure with atomic hydrogen sublattice.

## Results and discussion

**Synchrotron X-ray diffraction experiments**. The experiments were performed with a direct reaction of Ce and $H_2$ samples up to 159 GPa at room temperature. Compared with previously reported polyhydrides, the current reacted samples without laser heating are reliable enough to detect the X-ray diffraction (XRD) signals. Moreover, excess hydrogen in the sample chamber ensures the possible reaction of polyhydrides (Supplementary Fig. 1). We have identified the formation of Ce polyhydrides under high pressures. The newly synthesized phases were characterized by XRD from 3 to 159 GPa as shown in Fig. 1a. The evolution of $d$-spacing also proves the multiple phase transitions with increasing pressure (Supplementary Fig. 2). Because of the low-atomic scattering power of hydrogen, what we have directly determined are the positions of Ce atoms from XRD data. We have selected typical XRD patterns in Fig. 1b, and the results reveal dramatic changes in the formation of Ce polyhydrides.

At 3 GPa, just after the target sample loading into DAC, the diffraction pattern reveals that the sample crystallizes in a face-centered cubic (fcc) structure with the most possible space group $Fm$-$3m$ (Supplementary Fig. 3). This is reminiscent of the cubic structure of the element Ce at similar pressures, but with significantly large volume. With increasing pressure, the diffraction peaks broaden and shift abnormally, indicating a distortion in the fcc Ce sublattice (Supplementary Fig. 4). Above 33 GPa, distorted fcc phase gradually transfers to another cubic phase and the new cubic phase is stable up to 72 GPa. We observe an emergence of tetragonal phase at 62 GPa with a small diffraction peak. Note that the tetragonal phase can stabilize in a wide pressure range from 62 to 159 GPa. Above 80 GPa, a new phase with hexagonal symmetry appears and coexists with the tetragonal phase. Above 103 GPa, the crystal symmetry and diffraction patterns remain hexagonal symmetry, but its unit cell volume changes as discussed below. Currently, XRD patterns suggest that the original sample undergoes four phase transitions under high pressures. Upon decompression to ambient conditions, the hexagonal phase was recovered to initial fcc phase.

Our $P$–$V$ data points are plotted in Fig. 2 together with literature data. The present data are less scattered thanks to the improved volume accuracy. The $P$–$V$ data points are fitted with a third-order Birch–Murnaghan (BM) equation of state[24], yielding the zero pressure volume $V_0$, the bulk modulus $B_0$, and its pressure derivative $B_0'$. At 3 GPa, the volume of fcc phase is 41.80 $Å^3$/formula. Because of the very similar volumes between $CeH_2$ and $CeH_3$[25], we have identified fcc phase as $CeH_3$ rather than $CeH_2$, because $H_2$ is excessive in the chamber and the enthalpy calculations indicate that $CeH_3$ has a lower enthalpy value than $CeH_2$ (Supplementary Fig. 5). Above 72 GPa, the mixing curve of $Ce + 4/2\ H_2$ (here we use reported EOS of Ce[26] and $H_2$[27]) is in good agreement with our experimentally fitted compression curve of tetragonal phase, thus this phase is likely to have an $CeH_4$ stoichiometry. For the second cubic phase from 40 to 72 GPa, its volume is close to cubic $CeH_3$ at 40 GPa, however, the volume becomes close to tetragonal $CeH_4$ at 72 GPa. In this pressure range, we can also intuitively see that the compression curve of second cubic phase is approaching the ideal mixing curve of $Ce + 4/2\ H_2$ and connects the compression curves of $CeH_3$ and $CeH_4$ smoothly, indicating a process of hydrogen absorption from $CeH_3$ to $CeH_4$ (Supplementary Figs. 6 and 7). In the pressure range of 80–103 GPa, anomalous compressibility behavior of hexagonal phase is interpreted as an increase of the hydrogen concentration, corresponding to the ideal mixing curve of $Ce + 8/2\ H_2$. Above 103 GPa, the fitted compression curve of hexagonal phase lies above the ideal mixing curve of $Ce + 8/2\ H_2$ and below the ideal mixing curve of $Ce + 9/2\ H_2$. The calculation

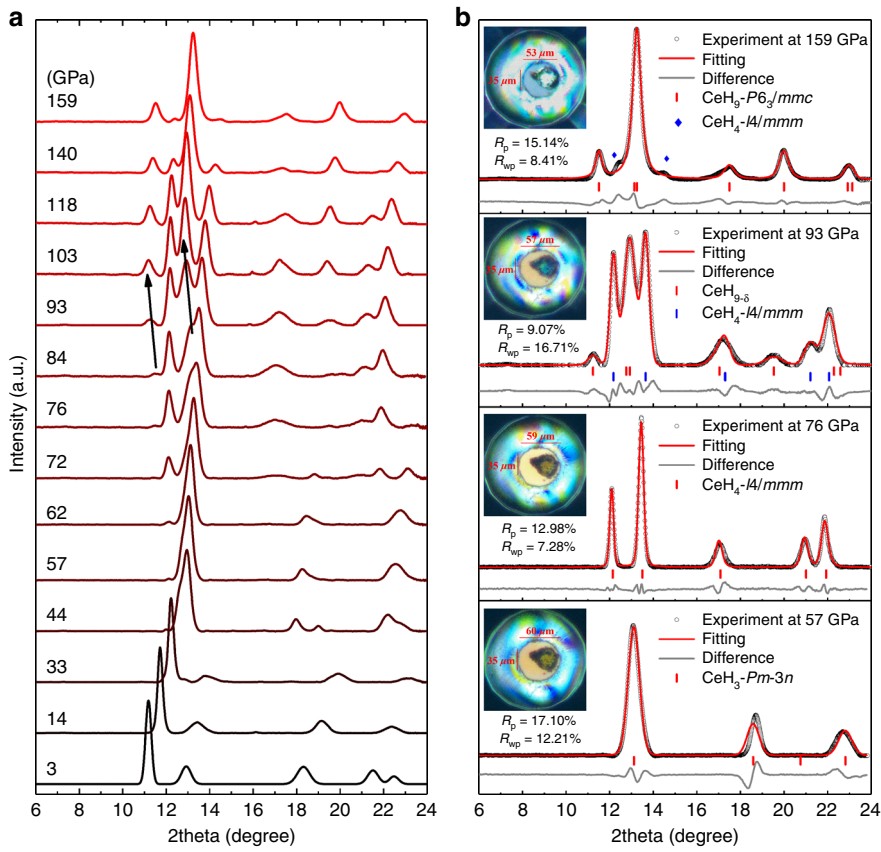

**Fig. 1** The evolution of XRD patterns of Ce polyhydrides at selected pressures. **a** Typical XRD patterns of Ce polyhydrides as a function of pressure upon cold-compression up to 159 GPa at room temperature. The black arrows show the abnormal trends of peaks with increasing pressure. **b** The Rietveld refinements of CeH$_3$-Pm-3n, CeH$_4$-I4/mmm and CeH$_9$-P6$_3$/mmc and Le Bail refinements of CeH$_{9-\delta}$, respectively at selected pressures. The incident wavelength was $\lambda = 0.6199$ Å. Excess solid hydrogen remained transparent at 159 GPa as shown in inserted photomicrographs

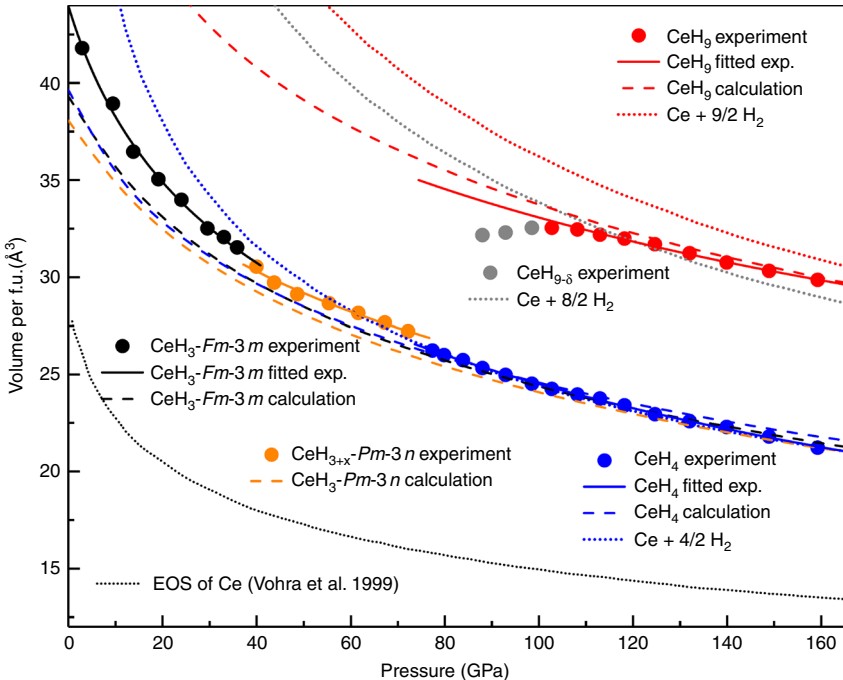

**Fig. 2** The volume per formula unit as a function of pressure for Ce polyhydrides. The solid points represent the experimental P–V data. The solid curves represent the BM equation fits of experimental results. The fitted results are $B_0 = 54.5$ (3.6) GPa, $B_0' = 4$ (fixed), and $V_0 = 44.0$ (0.5) Å$^3$ for CeH$_3$-Fm-3m; $B_0 = 106.0$ (13.4) GPa, $B_0' = 4$ (fixed), and $V_0 = 38.3$ (0.9) Å$^3$ for CeH$_{3+x}$; $B_0 = 95.6$ (2.5) GPa, $B_0' = 3.4$ (fixed), and $V_0 = 39.6$ (0.3) Å$^3$ for CeH$_4$-I4/mmm; $B_0 = 135.9$ (12.7) GPa, $B_0' = 4$ (fixed), and $V_0 = 47.4$ (1.0) Å$^3$ for CeH$_9$-P6$_3$/mmc. The dashed curves represent the calculated EOS. The dotted curves represent the volume of mixtures of unreacted Ce and H$_2$

of enthalpy in this pressure range shows that the hexagonal phase is in favor of CeH$_9$.

**Theoretical calculation of Ce polyhydrides.** The theoretical calculation is urgently required to determine the hydrogen positions and consequently the stoichiometry of Ce polyhydrides. Previous theoretical work has predicted five stoichiometries of Ce polyhydrides: CeH$_3$-*R-3m*, CeH$_4$-*I4/mmm*, CeH$_8$-*P6$_3$mc*, CeH$_9$-*P6$_3$/mmc*, and CeH$_{10}$-*Fm-3m*[21]. Inspired by the prediction on Ce polyhydrides, we did further detailed calculations on predicted structures of Ce polyhydrides to supplement our experimental results. Because Ce atoms with *f* electrons are an extremely delicate system for density functional theory (DFT) calculations, generalized gradient approximation (GGA) with Hubbard U correction was used for DFT calculations. We have done the structural searches at 0 K and 50, 100, and 150 GPa using CALYPSO and AIRSS methods[28,29]. We reproduced the three phases CeH$_4$-*I4/mmm*, CeH$_8$-*P6$_3$mc*, and CeH$_9$-*P6$_3$/mmc*, and proposed a new stable phase CeH$_3$-*Pm-3n*. As summarized in Supplementary Fig. 8, we have investigated the chemical stability of various Ce polyhydrides by calculating their formation enthalpies at pressures of 0, 50, 100, and 150 GPa, relative to the products of their dissociation into constituent. At 0 GPa, the formation enthalpies of all stoichiometries are positive, except CeH$_3$-*Pm-3n*. Both CeH$_4$-*I4/mmm* and CeH$_8$-*P6$_3$mc* phases lie on the convex hull at 50 GPa. As pressure increases up to 100 GPa, CeH$_9$-*P6$_3$/mmc* emerges, and CeH$_4$-*I4/mmm* is the most stable stoichiometry. At 150 GPa, CeH$_3$-*Pm-3n*, CeH$_4$-*I4/mmm*, and CeH$_9$-*P6$_3$/mmc* are energetically the most stable, falling on the convex hull. Besides, we investigated the dynamical stability of proposed CeH$_3$-*Pm-3n*, CeH$_4$-*I4/mmm*, CeH$_8$-*P6$_3$mc*, and CeH$_9$-*P6$_3$/mmc* by calculating their phonon dispersion curves, and found that all the phases are dynamically stable except CeH$_8$-*P6$_3$mc* (Supplementary Fig. 9).

**Crystal structures of Ce polyhydrides.** By the joint of experimental and theoretical results, the stoichiometries and crystal structure of compounds were estimated as follows: (i) XRD data give the possible phase transitions and symmetry and match the structures of prediction; (ii) the volume changes obtained from XRD data indicate the stoichiometry of new phases; (iii) theoretical calculations help us to finally determine the crystal structure and stoichiometry through the calculated enthalpy difference, pressure–volume relation and phase stability. Particularly, Rietveld refinement combines the calculated and experimental XRD results. The stoichiometries and crystal structures of Ce polyhydrides have been unraveled through the analysis of theoretical simulation and XRD experiments. As shown in Fig. 1b, excellent Rietveld refinements of experimental pattern profiles enable us to unambiguously determine the crystal structures of these polyhydrides. The experimental refined lattice parameters are summarized in Supplementary Figs. 10 and 11. At 3 GPa, the patterns of fcc phase match the known structure of CeH$_3$-*Fm-3m*. The second cubic phase matches our predicted CeH$_3$-*Pm-3n* structure and its crystal structure is presented in Supplementary Fig. 12. Nevertheless, the *P–V* curves demonstrate that CeH$_3$-*Pm-3n* would absorb hydrogen and become CeH$_{3+x}$ ($0.7 \leq x \leq 1.1$) as pressure increasing. The tetragonal phase matches the predicted CeH$_4$ with *I4/mmm* structure and its crystal structure is shown in Fig. 3a. In the pressure range of 80–103 GPa, although the predicted CeH$_9$-*P6$_3$/mmc* has been observed. The abnormal increase of *d*-spacing and lattice parameters indicates a significant expansion of volume (Fig. 2 and Supplementary Fig. 13), which is attributed to absorption of hydrogen in the process of forming CeH$_9$-*P6$_3$/mmc*. Because of the continuous expansion of volume, we assign the phase in this pressure range noted as CeH$_{9-\delta}$ ($\delta \leq 0.85$). Above 103 GPa, CeH$_9$-*P6$_3$/mmc* is successfully synthesized, with Wyckoff positions Ce: 2*d* (2/3, 1/3, 1/4) and H:2*b* (0, 0, 1/4), 4*f* (1/3, 2/3, 0.149), and 12*k* (0.156, 0.312, 0.062) shown in Supplementary Table 1. Most interestingly, CeH$_9$-*P6$_3$/mmc* presents a three-dimensional network in H sublattice. Figure 3b–d shows the crystal structure of CeH$_9$-*P6$_3$/mmc* and the H structure within CeH$_9$-*P6$_3$/mmc*. Each Ce atom is surrounded by 29 H atoms forming H$_{29}$ cage and there are 6 H$_4$ rings, 6 H$_5$ rings, and 6 H$_6$ rings in each H$_{29}$ cage. Contiguous H$_{29}$ cages share the rings and form an extended H network structure in three-dimensional space. Note that the *P–V* relation of these polyhydrides measured

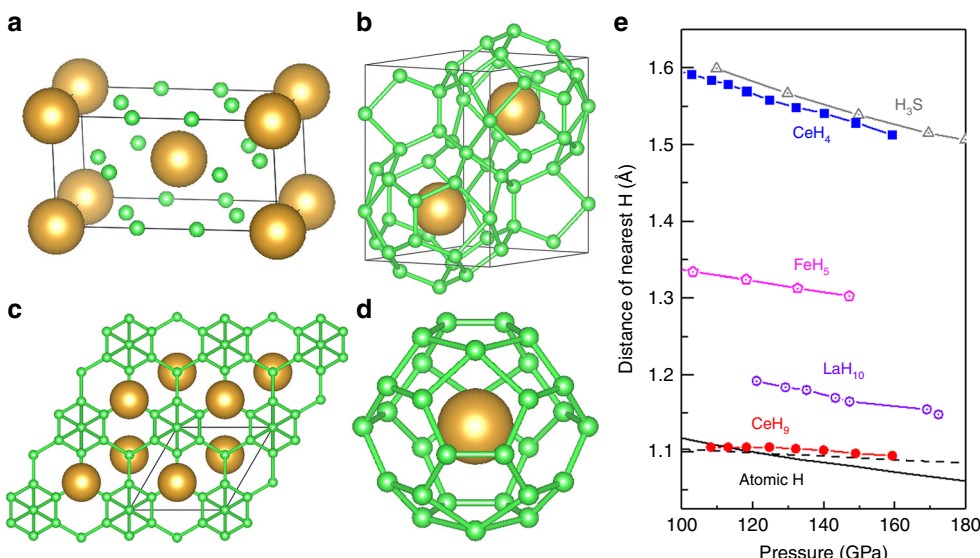

**Fig. 3** Crystal structure determination of Ce polyhydrides. **a** The crystal structure of CeH$_4$-*I4/mmm*. **b** The crystal structure of CeH$_9$-*P6$_3$/mmc*. **c** The extended H network in CeH$_9$-*P6$_3$/mmc*. **d** The H$_{29}$ cage in CeH$_9$-*P6$_3$/mmc*. Large brown and small green spheres represent Ce and H atoms, respectively. **e** The evolution of nearest-neighbor H–H distances in polyhydrides and atomic metallic H$_2$ as a function of pressure. These data were gained in previous work of H$_3$S[9], FeH$_5$[19], LaH$_{10}$[12], CeH$_4$-*I4/mmm*, and CeH$_9$-*P6$_3$/mmc* (this work) and the calculation of atomic metallic hydrogen (solid curve for ref. [19] and dash curve from ref. [12])

by the XRD are in excellent agreement with the theoretical calculation (Fig. 2), thus further confirming the stoichiometry. For the first time, the phase diagram of Ce–H system has been determined in experiment by the XRD patterns (Supplementary Fig. 14). The stable Ce polyhydrides in excess hydrogen have increasing stoichiometry under high pressure. Further discussion about stoichiometries is given in Supplementary Information.

Based on the above analysis, the polyhydrides especially $CeH_9$-$P6_3/mmc$ have been discovered merely through the cold-compression treatment. The present synthesized path facilitates further experimental characterization in contrast with other recently synthesized polyhydrides such as $LaH_{10}$ and $NaH_7$, which require laser-heating to overcome the energy barrier. Besides, we have also combined high-pressure and high-temperature conditions to trigger the formation of $CeH_9$-$P6_3/mmc$ and high-temperature is realized through laser-heating treatment. By means of Rietveld Refinement (Supplementary Fig. 15), we have confirmed the formation of $CeH_9$-$P6_3/mmc$ in these two synthetic routes—cold compression and high-temperature annealing.

**Atomic-like hydrogen sublattice realized via $CeH_9$-$P6_3/mmc$.** The discovered Ce polyhydrides invite us to reveal rich chemistry of hydrogen atom at elevated pressures. We compared the nearest-neighbor H–H distances and volume collapse per H atom in Ce polyhydrides with other reported polyhydrides and metallic hydrogen in Fig. 3e and Supplementary Fig. 16. Some polyhydrides consisted of $H_2$ or $H_3$ units, such as $Xe(H_2)_8$[30] and $NaH_7$[20], present undissociated H–H distances of about 0.75 Å. At 132 GPa, the nearest-neighbor H-H distance of $CeH_4$-$I4/mmm$ is 1.54 Å, which is slightly smaller than 1.56 Å in high-temperature superconducting phase $H_3S$. The nearest-neighbor H–H distance of $CeH_9$-$P6_3/mmc$ is closest to atomic metallic hydrogen (predicted $I4_1/amd$ structure[31]) with nearest-neighbor H–H distance of 1.09 Å at 132 GPa, which is also much lower than 1.2 Å in $LaH_{10}$[12], 1.32 Å in $FeH_5$[19] at the same pressure, indicating that Ce atoms play a more positive on chemical precompression. Thus, the evolution of nearest-neighbor H–H distance is almost close to atomic metal hydrogen, which may be realized by Ce atom doping.

We have calculated the electron localization function to figure out the bonding nature in the $CeH_9$-$P6_3/mmc$. Figure 4a shows that the lack of electron localization between Ce and H atoms indicates the absence of covalent bond characteristics and there is a weak electron localization (about 0.5–0.6) between H atoms forming a 3D network structure with $H_4$, $H_5$, and $H_6$ rings. In Fig. 4b, an analysis of the electronic band structure illustrates that $CeH_9$-$P6_3/mmc$ presents metallic character owing to overlap of the conduction and valence bands at the Fermi level. The projected electronic density of states shows that both H and Ce

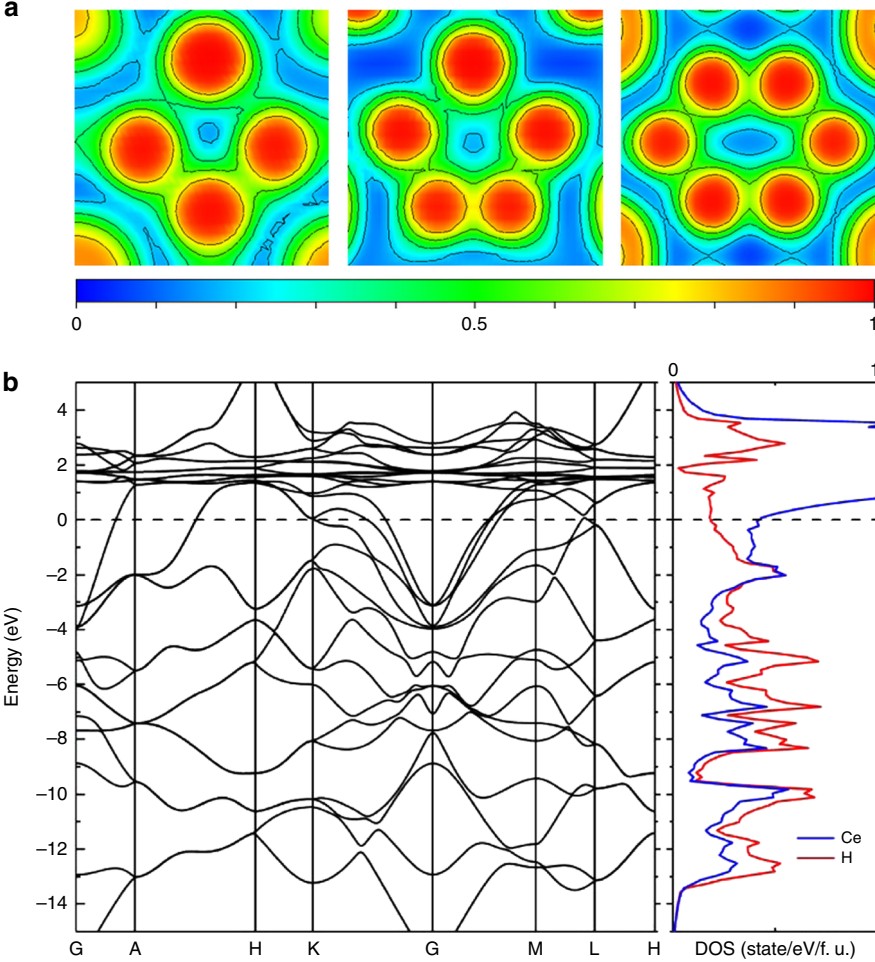

**Fig. 4** Calculated electronic properties of $CeH_9$-$P6_3/mmc$ at 100 GPa. **a** Plot of electron localization function reveals a weak electron localization (about 0.5–0.6) between H atoms and confirms the formation of $H_4$, $H_5$, and $H_6$ rings as the units of extended H network. **b** Electronic band structure and Partial density of electronic states of $CeH_9$-$P6_3/mmc$

make a sizable contribution to the density of states at the Fermi level (Fig. 4b and Supplementary Fig. 17). The density of electronic states of each phase illustrates that the contribution of hydrogen to the Fermi level would increase as stoichiometry and pressure increase (Supplementary Fig. 18), further confirming the indispensable role of hydrogen in these polyhydrides.

In conclusion, we have successfully synthesized a series of cerium polyhydrides by direct reaction of Ce and $H_2$ upon cold-compression up to 159 GPa. The Ce polyhydrides ($CeH_3$, $CeH_{3+x}$, $CeH_4$, $CeH_{9-\delta}$ and $CeH_9$) present an increase of hydrogen content as pressure increases. The formed $CeH_9$ has a unique clathrate-like structure consisting of $H_{29}$ cages surrounding Ce atoms occupying the hexagonal $P6_3/mmc$ symmetry. $CeH_9$ is also with the nearest-neighbor H–H distances closest to predictions for solid atomic metallic hydrogen in all synthesized hydrides. The electron localization function of $CeH_9$ indicates an ionic bonding between Ce and H atoms, and a weak electron localization between two H atoms, and the band structure confirms its metallic character. The density of electronic states calculations indicates significant contribution of H at the Fermi level. Our cold-compression experiment provides a facile route to potential superconductors in superhydrides. The discovery of $CeH_9$ with atomic-like hydrogen sublattice suggests a low-pressure route for bulk dense atomic hydrogen stabilized by other element atoms.

## Methods
**Experimental details**. We have carried out several experiments and the highest pressure reached 159 GPa. The diamonds were coated with 100 nm thick layer of $Al_2O_3$ to prevent diamond failure by hydrogen diffusion. Tungsten gaskets were also coated with a 100 nm gold to prevent the diffusion of hydrogen. The symmetric DACs were used to generate pressure. Ce powder with a purity of 99.99% was purchased from Alfa Aesar. All the sample of Ce were loaded and sealed in DACs inside glove box with Ar atmosphere (less than 0.01 ppm of oxygen and water) and the cells were transferred to gas-loading apparatus. The hydrogen was compressed to 0.17 GPa, then the cells were unclamped to allow the hydrogen to flow into the chamber and resealed. The Raman shift of diamond edge was used to calibrate the pressure[32].

The XRD experiments were performed at the 4W2 beamline of Beijing Synchrotron Radiation Facility (BSRF) and 15U1 beamline of Shanghai Synchrotron Radiation Facility (SSRF). The incident wavelength of X-ray was 0.6199 Å. The MAR image plate detectors were used to collect the diffraction patterns and the two-dimensional XRD images were radially integrated using DIOPTAS software[33], yielding intensity versus diffraction angle $2\theta$ patterns. Prior to measurement, a $CeO_2$ standard was used to calibrate the geometric parameters. The cold-compression experiments (3–159 GPa) were carried out at ambient temperature. The Ce polyhydrides would be synthesized only through cold-compression without laser-heating treatment. Additional laser-heating experiments were performed at 96 GPa and below 1500 K. The Reflex module in Materials Studio Program[34] and GSAS software[35] were utilized to index and refine the XRD patterns.

**Computational details**. First-principles calculations were performed with the pseudopotential plane-wave method based on DFT implemented in VASP code[36]. The GGA with the Perdew–Burke–Ernzerhof exchange-correlation functional was used in the calculation. The project-augmented wave method was adopted with valence electrons of $4f^15s^25p^65d^16s^2$ and $1s^1$ for Ce and H atoms, respectively. A plane-wave cutoff energy of 1000 eV was employed for norm-conserving pseudo-potentials and Brillouin zone sampling grids of spacing was $2\pi \times 0.03$ Å$^{-1}$. Phonon calculations were performed by a finite displacement method with the PHONOPY code[37]. The proposed structures were predicted through AIRSS[28] and CALYPSO[29] methods. To verify the correction of Hubbard U, an effective Hubbard parameter of 4.0 eV was added for the Ce $4f$ states in all calculations, which was used in previous calculation work of cerium hydrides[21]. The method of Methfessel–Paxton was used in the calculation of geometry optimization, band structure, electron localization function, phonon spectrum and the tetrahedron method with Blöchl corrections was used in DOS calculations.

## Data availability
All data supporting the findings of this study are included in this Article and its Supplementary Information, and are also available from the corresponding authors upon reasonable request. The source data underlying Figs. 1, 2, 3e, and 4b and Supplementary Figs. 2–8, 11, 12, 14–16, and 18 are provided as a Source Data file.

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

## Acknowledgements

The authors would like to thank professor Artem R. Oganov for many useful discussions, the Beijing Synchrotron Radiation Facility HP-Station 4W2, Shanghai Synchrotron Radiation Facility Beamline BL15U1 and its staff for allowing us to perform the XRD analyze. This work was supported by National Natural Science Foundation of China (Nos. 51572108, 51632002, 11504127, 11674122, 11574112, 11474127, and 11634004), the 111 Project (No. B12011), Program for Changjiang Scholars and Innovative Research Team in University (No. IRT_15R23) and National Found for Fostering Talents of basic Science (No. J1103202).

## Author contributions

X.H. and T.C. designed the experiments. X.L. and D.Z. prepared the cells and loaded the hydrogen. X.L. and Y.H. performed the XRD experiments. X.L. and X.H. analyzed the experimental data. C.J.P., H.X., Qu Z., and D.D. performed and analyzed the calculations. X.L., X.H., and T.C. wrote and revised the paper. Qi Z. and B.L. discussed the results and commented on the paper.

## Additional information

**Competing interests:** The authors declare no competing interests.

