## [Peer Review File · Nature Communications]

Reviewers' comments:

Reviewer #1 (Remarks to the Author):

The authors have revised the manuscript taking into account some comments from the Referee Reports.

The present version has been improved in several respects, however I think the following points should be considered before publication:

1) In the paper, the authors do not mention the recent observation of near-room-temperature superconductivity in lanthanum hydrides,

which has been reported by two groups:

> A. P. Drozdov, et al., cond-mat/1808.07039 and cond-mat/1812.01561.

> M. Somayazulu et al., PRL 122, 027001 (2019).

This discovery is highly relevant to the field of high-pressure hydrides, and even more to the specific topic of this paper, since LaH₁₀ and CeH₉ are structurally-related, and a suitable discussion should be included in the introduction, tuning down, if possible, the reference to small H-H distance, which is not particularly relevant to this system.

2) GGA + U: the authors have carried out GGA+U calculations of the equation of state, and report that with a reasonable U=4.0 eV, the agreement between theory and experiment has improved compared to standard GGA calculations.

It is reasonable to assume that "+U" corrections are expected to influence not only

the EOS, but all properties of Ce hydrides, since Ce f electrons are notoriously badly described in mean-field DFT.

In particular, the authors should recalculate using GGA+U two properties which are crucial for this manuscript:

- 1) the dynamical stability of the CeH₉ P63/mmc phase at 100 GPa.
- 2) The band structure and density of states in fig. 4b.

The first is crucial to solve the disagreement with , cond-mat/1805.0260, the second to determine the actual position of the Ce f states, and their actual contribution to the DOS at the Fermi level.

In the computational details, the author should also provide details on the k-space integration method (tetrahedron, smearing?)

- 3) The current abstract of the paper is very hard to read, in particular the last sentence. Before publication, it should be revised, removing again arguments on H-H distances, and mentioning the LaH₁₀ discovery.

Reviewer #1 (Remarks to the Author):

Comments: The authors have revised the manuscript taking into account some comments from the Referee Reports. The present version has been improved in several respects, however I think the following points should be considered before publication:

Author reply: Thank you very much for your valuable comments and suggestions. We have revised the manuscript (highlighted in red) according to your suggestions and comments. Hopefully we have addressed all your concerns. The following is our reply to all the comments.

1) In the paper, the authors do not mention the recent observation of near-room-temperature superconductivity in lanthanum hydrides, which has been reported by two groups:

> A. P. Drozdov, et al., cond-mat/1808.07039 and cond-mat/1812.01561.

> M. Somayazulu et al., PRL 122, 027001 (2019).

This discovery is highly relevant to the field of high-pressure hydrides, and even more to the specific topic of this paper, since LaH₁₀ and CeH₉ are structurally-related, and a suitable discussion should be included in the introduction, tuning down, if possible, the reference to small H-H distance, which is not particularly relevant to this system.

Author reply:

We have revised the introduction part according to your comments. We have cited the three works and added the discussion about observation of near-room-temperature superconductivity in lanthanum hydrides. Please see highlighted sentences in Line 15, Page 2 in the revised manuscript.

We have also added the discussion about the structural relation between LaH₁₀

and CeH₉ in the Introduction part. Please see Line 3, Page 3.

The discussion of small H-H distance has been deleted in abstract and introduction parts.

2) GGA + U: the authors have carried out GGA+U calculations of the equation of state, and report that with a reasonable $U=4.0$ eV, the agreement between theory and experiment has improved compared to standard GGA calculations.

It is reasonable to assume that "+U" corrections are expected to influence not only the EOS, but all properties of Ce hydrides, since Ce f electrons are notoriously badly described in mean-field DFT.

Author reply:

We are very sorry for the confusion of calculation details. The GGA+U calculations have been done on geometry optimization, band structure, DOS, and ELF in the manuscript. The dynamical stability of the synthesized phases has been recalculated with GGA+U (Fig. R1). "+U" corrections would not change the dynamical stability of synthesized phases.

Fig. R1 (a-d) Phonon dispersion curves for CeH₃-Pm-3n, CeH₄-I4/mmm, CeH₈-P6₃mc and CeH₉-P6₃/mmc at 50, 50, 80 and 100 GPa, respectively. Imaginary phonons observed in CeH₈-P6₃mc at 50 GPa.

In particular, the authors should recalculate using GGA+U two properties which are crucial for this manuscript:

- (1) the dynamical stability of the CeH_9 $P6_3/mmc$ phase at 100 GPa.
- (2) The band structure and density of states in fig. 4b.

The first is crucial to solve the disagreement with cond-mat/1805.0260, the second to determine the actual position of the Ce f states, and their actual contribution to the DOS at the Fermi level.

Author reply:

(1) We carried out GGA+U calculations of the dynamical stability of CeH_9 - $P6_3/mmc$ at 100 GPa, as shown in Fig. R2a. Compared with GGA calculation (Fig. R2b), "+U" corrections would not influence the dynamical stability of the CeH_9 .

Fig. R2 Phonon dispersion curves for CeH_9 - $P6_3/mmc$ with (a) GGA+U and (b) GGA.

We calculated dynamical stability of CeH_9 at 120 GPa and 150 GPa using GGA+U (Fig. R3), in order to compare with that in cond-mat/1805.0260, and we found dynamical stability is agree with their results at 150 GPa. However, our phonon results illustrate that there is no imaginary frequencies around H-K points in the phonon dispersion at 120 GPa and CeH_9 - $P6_3/mmc$ is dynamical stable.

Fig. R3 Phonon dispersion curves for CeH₉-P6₃/mmc with GGA+U at (a) 120 GPa and (b) 150 GPa.

(2) The band structure and density of states shown in Fig. 4b have already been the results of GGA+U calculations. We have shown the partial density of electronic states in Fig. R4. We found substantial contribution of Ce *f* electrons and H *s* electrons to the DOS at the Fermi level of CeH₉ with 31.38% by H and 57.61% by Ce 4*f* states.

Fig. R4 The partial density of electronic states of CeH₉ at 100 GPa.

In the computational details, the author should also provide details on the k-space integration method (tetrahedron, smearing?)

Author reply:

The method of Methfessel-Paxton ($ISMEAR = 1$, $SIGMA = 0.2$) was used in the calculation of geometry optimization, band structure, ELF and phonon spectrum and tetrahedron method with Blöchl corrections ($ISMEAR = -5$) was used in DOS calculations.

In the revised manuscript, we have described the GGA+U calculations in detail and added the k-space integration methods in the parts of discussion and computational details. Please see Line 5, Page 7 and Line 25, Page 13. We have added the discussion of PDOS and put Fig. R4 into Supplementary materials.

3) The current abstract of the paper is very hard to read, in particular the last sentence. Before publication, it should be revised, removing again arguments on H-H distances, and mentioning the LaH10 discovery.

Author reply: We have revised the abstract according to your comments.

REVIEWERS' COMMENTS:

Reviewer #1 (Remarks to the Author):

The authors have addressed all points raised in my previous reports, and I find their answers satisfactory.

I would recommend updating the arXiv Reference on LaH10 discovery to the published version: A. P. Drozdov et al., Nature 569, 528–531 (2019).

REVIEWERS' COMMENTS:

Reviewer #1 (Remarks to the Author):

The authors have addressed all points raised in my previous reports, and I find their answers satisfactory.

I would recommend updating the arXiv Reference on LaH10 discovery to the published version: A. P. Drozdov et al., Nature 569, 528–531 (2019).

Reply: Thank you for several round reviews. We have updated the arXiv Reference on LaH10 to the published version: A. P. Drozdov et al., Nature 569, 528–531 (2019).